# New Approaches in Heart Research: Prevention Instead of Cardiomyoplasty?

**DOI:** 10.3390/ijms24109017

**Published:** 2023-05-19

**Authors:** Ralf Gaebel, Cajetan Lang, Praveen Vasudevan, Larissa Lührs, Katherine Athayde Teixeira de Carvalho, Eltyeb Abdelwahid, Robert David

**Affiliations:** 1Department of Cardiac Surgery, Rostock University Medical Center, 18057 Rostock, Germanyrobert.david@med.uni-rostock.de (R.D.); 2Department of Life, Light & Matter, Interdisciplinary Faculty, Rostock University, 18059 Rostock, Germany; 3Advanced Therapy and Cellular Biotechnology in Regenerative Medicine Department, Pelé Pequeno Prίncipe Research Institute & Pequeno Prίncipe Faculties, Ave. Silva Jardim, P.O. Box 80240-020, Curitiba 1632, Brazil; 4Feinberg School of Medicine, Feinberg Cardiovascular Research Institute, Northwestern University, Chicago, IL 60611, USA

**Keywords:** cardiovascular disease, heart failure, obesity, physical exercise, prevention

## Abstract

Cardiovascular diseases are the leading cause of death in industrialized nations. Due to the high number of patients and expensive treatments, according to the Federal Statistical Office (2017) in Germany, cardiovascular diseases account for around 15% of total health costs. Advanced coronary artery disease is mainly the result of chronic disorders such as high blood pressure, diabetes, and dyslipidemia. In the modern obesogenic environment, many people are at greater risk of being overweight or obese. The hemodynamic load on the heart is influenced by extreme obesity, which often leads to myocardial infarction (MI), cardiac arrhythmias, and heart failure. In addition, obesity leads to a chronic inflammatory state and negatively affects the wound-healing process. It has been known for many years that lifestyle interventions such as exercise, healthy nutrition, and smoking cessation drastically reduce cardiovascular risk and have a preventive effect against disorders in the healing process. However, little is known about the underlying mechanisms, and there is significantly less high-quality evidence compared to pharmacological intervention studies. Due to the immense potential of prevention in heart research, the cardiologic societies are calling for research work to be intensified, from basic understanding to clinical application. The topicality and high relevance of this research area are also evident from the fact that in March 2018, a one-week conference on this topic with contributions from top international scientists took place as part of the renowned “Keystone Symposia” (“New Insights into the Biology of Exercise”). Consistent with the link between obesity, exercise, and cardiovascular disease, this review attempts to draw lessons from stem-cell transplantation and preventive exercise. The application of state-of-the-art techniques for transcriptome analysis has opened new avenues for tailoring targeted interventions to very individual risk factors.

## 1. Risk Factors for Developing Atherosclerosis

The underlying process that leads to the development of MI is atherosclerosis. Risk factors that favor the development of atherosclerosis include smoking, hyperlipidemia, arterial hypertension, diabetes mellitus, obesity, and a lack of exercise [1]. The endothelial cells are stimulated by one or more of these risk factors and express adhesion molecules and chemotactic molecules. The result is the recruitment of monocytes and T lymphocytes. Hyperlipidemia also leads to the deposition of lipids in the intima. In the course of atherosclerosis, the monocytes that differentiate into macrophages phagocytose these lipids and become so-called foam cells, characterized by a high proportion of intracellular lipids. The local leukocytes can further attract leukocytes by releasing inflammatory mediators and stimulating the smooth muscle cells of the media to migrate and proliferate. A fibrous cap toward the lumen covers the plaque. Collagen, produced by the smooth muscle cells, stabilizes the cap. The decisive factor in the development of coronary artery occlusion is not the size of the plaque but its vulnerability or stability. Local inflammatory processes weaken the plaque and its fibrous cap [2] by inhibiting the production of collagen I and III by smooth muscle cells via inflammation mediators such as interferon γ (IFN-γ). The foam cells produce matrix metalloproteinases, which break down the extracellular matrix [3]. The weakened fibrous cap can tear if it is mechanically overstressed. Two types are distinguished: deep ruptures with exposure of the thrombogenic lipid core and superficial erosion [4,5]. The result is the development of a thrombus. If this closes the vessel lumen, an MI occurs—only partial occlusion results in unstable angina pectoris [6].

Research into the genetic-physiological basis of myogenesis and adipogenesis is crucial to assessing the influence of risk factors that favor the development of atherosclerosis. Commonly, inbred mouse lines are used for most studies. However, human populations are genetically diverse, so inbred lines should be viewed critically when considering the genetic contribution to the disease. The use of non-inbred mice should be promoted.

## 2. Wound Healing after Acute MI

Myocardial healing in the first few weeks after acute MI is a highly dynamic and complex process that has been studied in great detail, especially in the mouse model. The biological processes occur in three characteristic phases in the hypoxic damaged murine myocardium. In the first 72 h, there is an intense inflammation dominated by infiltrating leukocytes, which clear away dead cells and the intercellular matrix. In the subsequent proliferative phase (up to approximately seven days after acute ischemia), the infarcted myocardium is replaced by granulation tissue. Fibroblasts differentiate into myofibroblasts and secrete an extracellular matrix traversed by a sprouting vascular network. In the maturation phase, fibroblasts undergo apoptosis, the angiogenetically fused vascular network recedes, and a collagen-rich scar develops [7]. In contrast, Ly-6C^low^ monocytes play a central role in the proliferative phase by stimulating fibroblasts, angiogenesis, and collagen formation [8]. Nahrendorf and colleagues hypothesized that two different entities of monocytes largely dominate these phases. Immunophenotypically, they are distinguished by high and low expression of the lymphocyte antigen Ly-6C. Ly-6C^high^ and Ly-6C^low^ monocytes are chemotactically directed into the myocardium via the cytokine gradients of CCR2 (CC chemokine receptor type 2) or CX3CR1 (CX3C chemokine receptor 1). The Ly-6C^high^ monocytes are phagocytic and proteolytically active cells that dominate the inflammatory phase.

The sprouting of blood vessels into the infarct area—angiogenesis—is an integral part of the healing myocardium, which is coordinated mainly by the cytokine environment of the inflammatory cells. Angiogenesis describes the de novo sprouting of capillary networks from endothelial cells in existing post-capillary venules [9]. Tissue ischemia is considered the trigger, which leads to the activation of HIF-1 (hypoxia-inducible factor) and consequently to the expression of pro-angiogenic factors such as VEGF (vascular endothelial growth factor). During the proliferative phase, it is a prerequisite for effective healing after MI and is mainly coordinated by monocytes and macrophages [10].

A skin lesion heals in a strictly organized sequence of the immune response, angiogenesis, and, ultimately, scarring. Wound healing following tissue damage is a highly conserved repair process similar to most humans and vertebrates’ organs [11,12]. The excellent accessibility of skin wounds enables an uncomplicated investigation of several points in time in the same model organism. With the help of the data on the healing of skin wounds, interesting conclusions can be drawn about the healing of the heart, which is much more difficult to access.

## 3. Impaired Wound Healing in the Case of Obesity

Impaired wound healing is an important health problem. Signs of aging and diseases such as stress and diabetes, as well as obesity, can lead to a chronic inflammatory state at a low level, which may disrupt wound healing. Obesity has a significant adverse impact on the wound-healing process. The latest findings indicate that aberrant inflammation of the wound site may be the cause of delayed healing [13]. Interventions that improve this effect are of particular interest [13]. Rats fed a high-fat diet of 30% total calories for 15 weeks retained a significantly higher percentage of the initial wound area after wounding after 14 and 21 days than non-obese controls [14]. The re-epithelialization of the wound area was also impaired in animals fed high-fat food. The control group showed more significant infiltration of fibroblasts 21 days after the wounding than their obese conspecifics. In an experimental model of abdominal laparotomy with induction of a skin wound and scarring, rats that were given a high-fat diet for four months developed a wound with reduced tear resistance compared to non-obese control animals [15]. In a similar model for laparotomy wound healing in genetically obese sugar rats, Xing and colleagues also observed a reduced tear resistance of the larger wound 28 days after the operation [16].

Interestingly, a comparative study showed that high-fat diet (HFD) mice did not develop any chronic wounds in contrast to genetically diabetic (ob/ob) mice. However, healing was delayed as a result [17]. While ob/ob and db/db mouse models for studies of the healing mechanisms of chronic wounds, such as diabetic leg ulcers, are preferred, HFD mice may be much better suited for investigating delayed healing of acute wounds, such as those caused by bariatric or other surgery in obese patients.

Wound healing consists of integrated and overlapping phases of hemostasis/coagulation, inflammation, proliferation, and dissolution/remodeling. These phases must occur at defined times and last for a certain period; otherwise, pathological wound healing occurs [18]. Chronic wounds, such as those that occur in type II diabetes, cannot go through these discrete stages due to dysregulated inflammation during the inflammatory phase of wound healing [19,20,21,22,23]. A common characteristic of these poorly healing wounds is a minor initial immune response to an injury. In normal wound healing, the early innate inflammatory reaction is decisive for establishing the healing cascade [24,25,26]. During the first part of the inflammatory phase of wound healing, macrophages exist in an inflammatory phenotype, releasing inflammatory cytokines and mediators, recruiting additional leukocytes, and promoting tissue and pathogen destruction [26]. After this early inflammatory phase with normal wound healing, these macrophages change their phenotype and secrete anti-inflammatory mediators and growth factors to promote tissue repair and wound dissolution [26]. Studies in diabetic mouse models have shown that this critical early macrophage-mediated inflammatory response is impaired in diabetic wounds [24,25]. In addition, the pro-inflammatory to anti-inflammatory macrophage phenotype is impaired in several diabetes models in which a persistent hyper-inflammatory macrophage phenotype occurs [19,27,28]. The gradual change in the wound macrophage response from pro-inflammatory to anti-inflammatory is a key component of natural healing and is necessary for effective wound closure [28]. The ability to fully understand and control the initiation and resolution of inflammation in wound macrophages is critical to advancing the field of wound healing. It is currently unknown what explicitly drives changes in the phenotype of wound macrophages during healing [29,30,31,32]. Therefore, studying the molecular mechanisms underlying macrophage plasticity in wounds is needed to address the pathology observed in diabetes.

## 4. The Evolutionary Gene Chip

The human predisposition to obesity results from the interaction between biology and culture over the course of human evolution. Hominid ancestors and modern humans were more often confronted with food shortages on the one hand and had to be very physically active on the other [33]. Therefore, as mammals, humans have developed the ability to store body fat. Simultaneously with encephalization, humans developed sophisticated and complex genetic and physiological systems to protect themselves from hunger and to defend stored body fat. Both genes and cultural traits that were adaptive due to past food shortages now play a role in the etiology of maladaptive obesity in adults [34]. Since in most of human history, all food reserves were scarce, during evolution, gene variations were favored that allowed effective energy storage in fat deposits. This fact was crucial for women (who to this day have larger fat stores than their male counterparts), as they had to be able to breastfeed children even during periods of hunger. This ability to store calories increased the chances of reproduction and survival when food was scarce, which is why the corresponding gene variants prevailed. The former advantage only led to obesity since energy in the form of food was abundant and available at all times. At the same time, the physical activity of humankind has been drastically reduced in the post-industrialized world. The human body is not adapted to these new conditions. In the modern obesogenic environment, those individuals with the appropriate combination of energy-saving genes from their ancestors are at greater risk of being overweight or obese and the associated chronic consequences such as high blood pressure, diabetes, dyslipidemia, MI, and heart failure. It is known that extreme obesity affects the structure and function of the heart in terms of hemodynamic load, altered remodeling of the left ventricle (LV), and impaired ventricular function, leading to heart failure [35]. In at least four prospective community-level studies, the body mass index (BMI) gradually predicted the risk of heart failure beyond the known risk factors. The incidence of overweight and obesity in the western world has increased dramatically. In the United States, two-thirds of the adult population is overweight, and one-third is obese [36].

## 5. The Influence of Obesity on Infarct Healing

Paradoxically, obesity has been protective shortly after MI when associated with less comorbidity [37]. Nevertheless, the risk of recurrent MI and long-term adverse results increased in patients with a BMI > 25 kg/m^2^ [38]. In the Framingham Heart Study [39], a connection was established between BMI and the risk of developing heart failure. Obesity-related disorders are linked to the undesirable consequences of ischemic heart disease, although the mechanisms responsible are marginally defined. In addition, Nam and colleagues reported a higher risk of wound complications with infra-inguinal venous bypass grafting in obese patients [40]. Against the backdrop that an overweight-related chronic inflammatory condition generally worsens wound healing, it is urgently necessary to assess the influence of obesity on an MI’s healing process and explain it pathophysiologically. Characterizing a mouse model for obesity that fulfills many aspects commonly observed in human obesity enables a detailed study of the adverse cardiovascular effects of obesity at the molecular level. A major limitation of genetically obese mouse models for investigating inflammation and subsequent remodeling after infarction is the altered immune system in these animals, which may influence the remodeling process when myocardial ischemia/reperfusion damage develops [41]. Thakker and colleagues used a diet-related obesity mouse model to study the physiological effects of obesity and the inflammatory and healing responses of diet-related obese (DIO) mice after injury from myocardial ischemia and reperfusion [41]. After 24 h of reperfusion, they found that chemokines’ expression in lean mice’s infarcts had almost decreased to sham levels. Meanwhile, obese mice showed an unfavorable reaction regarding chemokines increased and prolonged expression. The profile of cytokine expression also changed in obese mice after MI. The prolonged expression of IL-6 showed persistent inflammation and, at the same time, an anti-inflammatory effect of IL-10. After 72 h, the macrophage density in the granulation tissue was significantly increased in DIO mice compared to lean animals, while the neutrophil density was reduced [41]. Phenotypically modified fibroblasts, or myofibroblasts, play a central role in fibrosis progression due to their ability to produce procollagens [42,43]. The collagen deposition within the scar is critical to maintaining the tensile strength of the myocardium and preventing rupture [44]. The density of myofibroblasts was not significantly different in the infarcts of lean and obese mice after 72 h of reperfusion. However, after seven days of reperfusion, obese mice had significantly reduced replacement collagen in the scar, which can contribute to adverse remodeling. DIO mice developed hyperinsulinemia, insulin resistance, and liver steatosis with significant ectopic lipid deposition in the heart and cardiac hypertrophy without significant changes in blood pressure [41]. Lipid deposition determined by Oil Red O staining was found in sections of the LV-free wall in a group of overweight (BMI > 30 kg/m^2^) people with diabetes and non-ischemic heart disease [45]. It is unclear whether intramyocardial lipid deposition is a feature of human heart failure. However, insulin resistance and diabetes are often associated with lipid deposition in tissues such as the liver and skeletal muscle [46,47,48]. It is generally believed that lipotoxicity, defined as tissue dysfunction induced by lipid deposition in non-obese tissues, is a cause of the development of insulin resistance. In addition, cardiac fatty acid metabolism disorders in animal models of diabetes and obesity lead to an ectopic accumulation of cardiac lipids [41]. Thus, Thakker and colleagues hypothesized that insulin resistance and increased circulating non-esterified free fatty acids in diet-related obesity would also lead to lipid accumulation in the heart. The results impacted the hearts of animals with diet-related obesity, as lipid species can potentially produce toxic metabolites that can harm cardiomyocytes, especially after an injury, and may impair cardiac function [41]. Zhou and colleagues showed cardiac lipid accumulation was associated with contractile dysfunction in obese rats with diabetic sugar fat [49]. To differentiate the contributions of global metabolic defects from those of the accumulation of cardiac lipids to the development of cardiomyopathies, several researchers in recent years have developed transgenic mice in which the intake of fatty acids exceeds the consumption of cardiac fatty acids. The cardiac-specific overexpression of acyl-CoA synthetase-1 [50], peroxisome proliferator-activated receptor-α or glycosylphosphatidylinositol-anchored lipoprotein lipase leads to lipid accumulation in the myocardium, which is associated with systolic ventricular dysfunction. Transgenic mice with cardiac overexpression of fatty acid transport protein-1 also import free fatty acids that exceed their usable capacity and show diastolic ventricular dysfunction [50,51,52,53]. Since the heart is highly dependent on fatty acid metabolism for its urgent energy needs, the balance between the intake and use of fatty acids is strictly regulated. Myocardial lipid estimates by magnetic resonance spectroscopy suggest that obese individuals (BMI > 30 kg/m^2^) with signs of impaired contractile function have abnormally high levels of triglycerides in the heart [54]. In addition to triglyceride deposition in the heart, DIO mice in the study by Thakker and colleagues had elevated levels of free fatty acids, especially saturated fatty acids, and arachidonic acid, which are proapoptotic and proinflammatory [41]. Elevated fatty acid levels are not only causally related to the development of insulin resistance. However, they are also closely linked to inflammatory pathways by activating genes that respond to κB [55,56].

## 6. Regeneration of Lost Heart Tissue Using Stem Cells—Lessons from Cardiac Cell Therapy

The data from the Federal Statistical Office show that chronic ischemic heart disease and acute MI caused more than 35% of all deaths in Germany in 2017, in most cases due to advanced coronary artery disease. In addition to hemodynamic, metabolic, and biochemical changes, structural damage to the heart tissue also affects the electrophysiology of the heart. Such damage is most common after acute MI and results in increased arrhythmogenicity in the heart. Arrhythmias with harmful prognostic properties lead to increased therapy-relevant mortality [57], whereby the probability of life-threatening cardiac arrhythmias increases with infarct size [58]. Initially, this arrhythmogenicity was based on an increased extracellular potassium concentration. As a result, an inhomogeneous distribution of potassium is observed, which leads to a modulation of excitability, spontaneous activation by Purkinje fibers, and the development of ventricular tachycardias [59]. In addition, increased sympathetic activity via anaerobic glycolysis, the formation of free fatty acids, and increased cytosolic calcium concentration led to a variation in the length of the action potential and repolarization. Free fatty acids have also been found to promote ventricular fibrillation [60].

Further changes are observed in the connexons. The dephosphorylation breaks the coupling of the cells of connexin 43, and the physiological conduction of excitation at the gap junctions is disturbed [61]. While this change within the scar has an anti-arrhythmogenic effect due to wave interruptions, there is an arrhythmogenic tendency at the border zone of the infarct area with fragmentation of the electromagnetic waves [59]. In addition, the induction of cardiac arrhythmias by coronary spasms after reperfusion of the damaged myocardium was observed [62].

However, while mortality from acute MI has decreased significantly since 1980, chronic ischemic heart disease as a secondary disease has been at the top of the list of the most common causes of death since 1992. In the early stages of the disease, pharmacological interventions can prevent progression and, if adherence to therapy is optimal, even reduce coronary stenoses [63]. Early revascularization can minimize myocardial damage, but every heart attack leaves behind irreversible damage. Endogenous remodeling and repair mechanisms subsequently lead to so-called ventricular remodeling, which is characterized by fibrosis, dilation, and hypertrophy of the ventricular myocardium and leads to the development of heart failure. The pathophysiological basis for this is reducing end-diastolic pressure via inhibiting the renin–angiotensin–aldosterone and sympathetic nervous systems. The only established therapies to reduce ventricular remodeling long term are neurohumoral pharmacological interventions with drugs such as angiotensin-converting enzyme inhibitors, beta-blockers, and spironolactone.

Due to the lack of causal treatment options, the last 30 years of cardiovascular research in cell transplantation have been shaped by the euphoric hope that lost heart tissue could be replaced with stem cells. Initially, the focus was on the intramyocardial transplantation of adult stem cells, which can promote specific cardiac functions and follow cardiovascular differentiation pathways but are safe to use. The therapeutic effects on the intercellular communication of CD271^+^ mesenchymal stem cells with surrounding cardiomyocytes were described, which trigger the intrinsic program of the cardiogenic lineage specification of the mesenchymal stem cells [64]. These were both precultured and primarily isolated human adult stem cells. Furthermore, we were able to show that CD271^+^ mesenchymal stem cells have no arrhythmogenic properties and are even a suitable cell type to prevent arrhythmias after MI [65].

In the last 20 years, different cell populations have been investigated concerning their therapeutic effects on ischemically damaged myocardium. Almost all cell preparations had a measurable positive effect on improving pump function. To obtain an overview of the enormously extensive research field and to summarize and systematically evaluate the countless stem-cell experiments in the mouse model for the therapy of acute MI, meta-analyses were started in this area [66] (Figure 1). Our study showed that cardiac stem-cell therapies resulted in an improvement in left ventricular ejection fraction of approximately 9%. Subgroup analysis of the data revealed that cardiovascular cells (among the non-pluripotent cells) had the most significant effect [67]. Based on this, in our experiments on cardiac stem-cell therapy, differentiated cardiovascular progenitor cells were transplanted from embryonic stem cells. We established MRI for volumetric measurements after a MI to measure the effect of regenerative therapies on cardiac pump function. Furthermore, we established the tracer 18F-NODAGA-RGD in small animal PET for quantifying Integrin α_V_β_3_ after MI [68]. It was able to image and quantify the inflammation after MI and the therapeutic effect of stem-cell therapy with cardiovascular progenitor cells by using the tracer 18F-FDG [69]. In connection with the transplantation of foreign cells after MI, we dealt with changes in the inflammatory response in the heart [69,70]. The unpublished studies demonstrate in a comparative experiment between C57BL/6 vs. immunodeficient RAG2del (no mature T or B cells) mice that the immune response plays a mediating role in favor of cell transplantation [71].

In addition, we addressed the regenerative effectiveness of hematopoietic CD133^+^ bone marrow stem cells, which is mainly based on the regulation of angiogenesis [64,72,73,74,75]. In the randomized, double-blind, placebo-controlled, multicenter phase 3 PERFECT study (intramyocardial CD133^+^ stem-cell transplantation/coronary artery bypass surgery) with patients after MI and coronary heart disease, we observed a preoperative signature of circulating bone marrow stem cells and endothelial progenitor cell parameters in the peripheral blood that correlated significantly with the postoperative response of myocardial regeneration [72]. Now, for the first time, we have found a significant feature of RNA gene expression that is characteristic of a reduced (non-responder) or increased (responder) proliferation reaction of bone marrow stem cells to angiogenesis [75]. Using an integrative algorithm (machine learning) for gene expression enabled us to identify patient-specific disorders that led to an individualized change in path mechanisms and targeted treatment for the first time. This independent machine-learning-based selection of non-responder/responder differentiation factors indicates myocardial repair through the modified expression of signal and adapter proteins. An exciting finding emerges from the results of cardiac stem-cell therapies in clinical studies: the healing process of the ischemically damaged myocardium can be modulated, and the extent of the myocardial damage can be reduced in this way. In addition, it has been noticed in clinical studies that, regardless of the therapeutic intervention, patients can be divided into so-called responders, i.e., patients whose left ventricular ejection fraction had improved by >5% 180 days after the bypass operation, and non-responders, whereby the authors make the hypothesis that the non-responsiveness is based on disturbed angiogenesis, which is caused by a dysfunctional bone marrow response [72] (Figure 2). In the setting of acute MI, angiogenesis is an integral part of the healing process, so the idea is that the plasticity of the healing capacity of an organism per se largely determines the extent of cardiac regeneration.

The successes of the phase III studies are relatively limited despite the enormous research effort; thus, the future of the research field remains unclear. Furthermore, to develop optimal stem-cell therapies, research is currently focused on the healing processes and their in vivo imaging and functional cardiac parameters after an MI. First, the fate of intramyocardial transplanted stem cells should be followed in vivo. By labeling transplanted murine embryonic stem cells with ^18^F-FDG, the biodistribution of the cells could be recorded and quantified in vivo in the first 2 h after injection into the heart [76,77]. Overexpressing the reporter protein hNIS (human sodium iodide synthase) in murine embryonic stem cells could visualize transplanted cells over a period of 6 weeks using ^124^I-PET [68].

## 7. Prevention in Cardiology in the Focus of Recognized Specialist Societies

Cardiac stem-cell therapies have been extensively studied over the past 20 years, but the results of almost all phase III clinical trials correlate only marginally with the promising preclinical data. In a position paper of the European Society for Cardiology (ESC) on the subject of cardiac stem-cell therapies [78], the summary begins with the words, “The early promise of cell therapy has not yet been fulfilled”. A possible and, at the same time, obvious explanation is the big difference between young healthy mice as a frequently used preclinical model and elderly sick patients concerning the healing capacities of the respective organisms and the quality of the respective stem-cell populations [79]. As a result, no single-cell therapy has been approved for treating heart failure or MI; there needs to be a recommendation in the guidelines of recognized specialist societies [79]. Despite major investments, the therapy of acute MI and heart failure has unfortunately not improved significantly, thanks to this research area. Against the background of the increasing prevalence of cardiovascular diseases, the associated suffering of patients, and the rising costs of the resulting health care, new approaches are urgently needed.

Coronary artery disease is largely caused by modifiable behaviors such as physical inactivity, an unhealthy diet, and smoking [80]. The incidence of complications from coronary artery disease could be massively reduced through targeted lifestyle interventions such as sporting activities. The cardiological societies have long agreed on the cardioprotective effect of a healthy lifestyle. Essential components are a healthy diet, non-smoking, normal body weight, and physical activity [81]. Corresponding recommendations are now part of the ESC guidelines for treating coronary heart disease [82]. However, a large part of the data comes from epidemiological studies, while there are few randomized and controlled prospective studies compared to pharmacological intervention studies [81]. The European Association of Preventive Cardiology has shown the ESC in a current position paper the gaps in cardiological prevention research and defined research goals for the coming years [83]. A translational concept is proposed to understand lifestyle interventions from the pathophysiological mechanisms in the animal model to the clinical manifestation of the respective disease [81].

The recommendations of the current ESC prevention guideline form a new basis for joint decision-making by doctor and patient in the selection of preventive measures based on individual patient characteristics. The therapy goals can be individually adapted in a step-by-step procedure. While BMI, waist circumference, and waist-to-hip ratio do not improve the prediction of cardiovascular disease risk, a comprehensive threat assessment should be considered in overweight/obese individuals. The classic BMI categories are increasingly being disaggregated to focus on waist circumference, as this is a marker of central obesity and is strongly associated with the development of CHD and diabetes. The current prevention goal is weight reduction to a normal weight (BMI < 25 kg/m^2^) and a waist circumference of ≤94 cm for men and ≤80 cm for women. To improve the risk profile of cardiovascular disease and reduce the threat of type 2 diabetes, overweight and obese people are advised to aim for weight reduction. Although a number of short-term diets are effective for weight loss, a healthy diet should be maintained over a longer period of time in view of the risk of cardiovascular disease. If lifestyle changes do not result in sustained weight loss, bariatric surgery and SGLT2 inhibitors and GLP-1RA to reduce future cardiovascular disease may be considered in high-risk obese patients (BMI > 40). The use of SGLT2 inhibitors and GLP-1RA is recommended for patients with coronary heart disease because of their proven prognostic benefits [84].

In addition, aspects of the current ESC guideline also include the use of a new risk algorithm based on cardiovascular morbidity and an adjusted risk calculation for patients with established atherosclerotic cardiovascular disease, obesity, and hypertension. For example, regular screening for sleep disorders is indicated. To reduce body weight and prevent or slow weight gain, lifestyle modifications, including smoking cessation, a low-fat, high-fiber diet, aerobic and resistance training, and reducing energy intake, are recommended for people with diabetes. In newly diagnosed diabetes mellitus, a hypocaloric diet with aggressive weight reduction can lead to remission of the disease. To this end, a low-calorie diet should be considered early after diagnosis, followed by a period of food reintroduction and a period of weight maintenance [84].

The trend towards overweight and obesity is currently pointing upward. Approximately 8% of all deaths worldwide are attributed to obesity, with a meta-analysis from 2017 showing that even moderate weight loss reduces all-cause mortality by 18% [85]. Effective prevention programs and rehabilitation measures in the case of manifest cardiovascular diseases mean that lifestyle modifications as well as drug prevention can be implemented in the long term [86,87]. The significant prognostic benefit of training-based multidisciplinary cardiac rehabilitation is also observed in meta-analyses [88].

In this way, a basis can be established to identify targeting mechanisms for biomarkers that represent the effect of personalized lifestyle interventions tailored to individual risk factors.

## 8. Influence of Endurance Training on the Healing of the Heart and Skin

Interventions that can accelerate the chances of healing for those with slow or non-healing wounds are critical. Studies show that exercise training tends to reduce inflammation in obesity, regardless of whether human or animal models are used [89,90]. Given the significant effects of exercise on inflammation seen in obesity, exercise should be able to reduce aberrant inflammation in wound tissue in obese individuals and thereby improve delays in wound healing. Emery and colleagues inflicted skin injuries on older adults one month after starting exercise [91]. Wound healing was then assessed until the wound was closed entirely. The athletes’ wounds healed significantly faster at 29 days compared to 39 days for seated subjects. No molecular measurements of wound status were made during the study. Therefore, no accurate conclusions can be drawn about the mechanisms behind these changes [91]. In a mouse model, Keylock and colleagues also found that physical activity significantly accelerated healing compared to the healing rate in the sedentary controls [92]. Interestingly, the main differences in the healing rate occurred between 1 and 5 days after the wounding, indicating the effect of exercise during an early period of so-called inflammatory healing. Therefore, the researchers assessed the inflammatory status of the wound using gene and protein expression of selected proinflammatory cytokines and chemokines and compared the trained mice with control animals. TNF-α was decreased on days 3 and 5 after the exercise wound, with an almost significant decrease also occurring on day one after the wound. MCP-1 (monocyte chemotactic protein) was also reduced during exercise on days 1 and 3 after the wounding. The data from this study suggest that exercise accelerates the rate of wound healing in aged mice and is associated with decreased inflammation. However, inflammatory cell infiltration does not change significantly [92].

Exercise is a relative intervention strategy that can be used clinically to prevent or treat disorders in the healing process. Little is known about the mechanisms by which movement accelerates healing. In addition, clinical studies on obese people are needed to determine whether the results can be transferred from obese animal models. We use the non-inbred mouse line DU6, established over 146 generations by phenotype selection for a high body mass 42 days after birth [93,94], as well as the Fzt:DU based on the identical outbred herd as wild type-control. We recently provided significant evidence at the single-cell transcriptome level for using such non-inbred mouse lines in studying myogenesis and adipogenesis. For this purpose, we isolated cores from whole hearts of adult mice of the same age, both from an inbred strain (C57BL/6) and from the outbred strain Fzt:DU [95]. In a comparative study between the inbred strain C57BL/6 and the outbred strain Fzt:DU using single-nucleus RNA sequencing, we were able to show the proportion of cardiomyocytes in the C57BL/6 strain is higher than in Fzt:DU mice (Figure 3) [96].

In contrast, there are twice as many endothelial cells in the Fzt:DU strain as in C57BL/6 mice [96]. Among these endothelial cells, we found a population with markers related to cardiomyocyte functions (e.g., Gja1, Atp2a2, Ttn, Ryr2, and Myh6). Supplementary RNA kinetic analyses used information about nascent mRNA as a predictor for future cell states and enabled the investigation of transcription kinetics [97]. The results support the idea that this population is in the process of transdifferentiation from an endothelial cell-like phenotype to a cardiomyocyte-like phenotype, which earlier studies confirmed [98]. In contrast, C57BL/6 mice lack this intermediate cell population [96]. In addition, we were able to show in this context that the threshold value of 5% customarily used in single-cell sequencing protocols for mitochondrial transcripts in the heart introduces an unacceptable bias, which leads to the exclusion of the majority of cardiomyocytes and especially pacemaker cells [99].

The protective effect of endurance training on the cardiovascular system has long been known, but the underlying molecular mechanisms are only partially understood [100]. In this regard, it was observed in mice and rats after experimentally induced MI that, compared to a lack of physical training, pre-infarct endurance training leads to improved cardiac pump function [101,102,103]. Calvert and colleagues postulated stimulation of β3 receptors and increased storage of nitric oxide metabolites in the heart as possible mechanisms [104]. The study of Freimann et al. (2005) showed at the transcriptional level post-infarction in mice that pre-infarct training leads to a less pronounced shift from aerobic glycolysis to anaerobic glycolysis [101]. In other studies, training resulted in a reduction in scar size after MI. It could be shown that pro-inflammatory cytokines such as TNF-α (tumor necrosis factor α) and IL-6 are expressed in a reduced manner [105] or that increased vascular density and VEGF expression occur in the infarct area [75].

Potential starting points for deciphering these mechanisms arise from new findings in the area of wound healing from injuries to the skin. Due to the preserved healing processes, the basic concepts should be transferable to myocardial healing [12]. The phenomenon observed in elderly patients of poorer wound healing than in juvenile organisms, associated with an altered immune response, is also reflected in old mice [106]. Macrophages and monocytes are central mediators of inflammation and angiogenesis in healing processes [107]. There is increasing evidence that impaired macrophage function and polarization in old age are essential components of impaired wound healing [106].

Interestingly, there is also increasing evidence that the disturbed inflammation profile and, thus, the skin healing capacity can be improved through endurance training [91,92]. That occurs through reprogramming the macrophages into the more juvenile functional type [106]. In lower vertebrates, such as zebrafish and some newts, the heart can at least predominantly regenerate after an injury [12], but also in neonatal mice, where the coordination of the inflammation in the healing process by other macrophage and monocyte populations plays an essential role [10]. Modulating the monocytic response toward the neonatal phenotype is a promising therapeutic option. In the context of the DU6 mouse line as a model for high-body-mass individuals, it would be of great interest whether these obese mice show a significantly altered inflammatory profile with regard to the monocyte and macrophage subpopulations compared to genetically unaffected animals.

At present, only initial data on the mechanisms underlying the cardioprotective effects of endurance training exists. A targeted search for possible biomarkers to measure the effectiveness of the lifestyle intervention sport—as required by the ESC—could clarify to what extent obesity plays a significant role during the healing process after MI due to the change in the cellular inflammation profile. If the individualized pathomechanisms identified in the mouse correlate with specific data from heart attack patients, the healing process after an MI could be significantly improved based on patient-specific diagnosis and targeted prediction, including treatment.

## 9. Conclusions

After intensive research into regenerative concepts such as stem-cell therapy over the last 20 years, the healing potential remains relatively limited despite the enormous research effort. Preventive approaches to activate endogenous repair mechanisms and, thus, on the one hand, minimize the extent of damage caused by tissue ischemia and, on the other hand, improve wound healing are therefore of great interest. In the search for alternative approaches to minimize the extent of functional damage, the improvement of endogenous healing mechanisms is more important than ever. The research groups are increasingly concentrating on the connection between the chronic inflammatory state of the heart and its associated negative impact on the wound healing process. Although the cardiological societies have long agreed on the cardioprotective effect of a healthy lifestyle, there still needs to be more understanding of a real translational lifestyle intervention concept, from the pathophysiological mechanisms in animal models to the clinical manifestation of the respective disease. The aim should be to identify mechanisms for biomarkers that represent the effect of personalized lifestyle interventions tailored to individual risk factors.

## Figures and Tables

**Figure 1 ijms-24-09017-f001:**
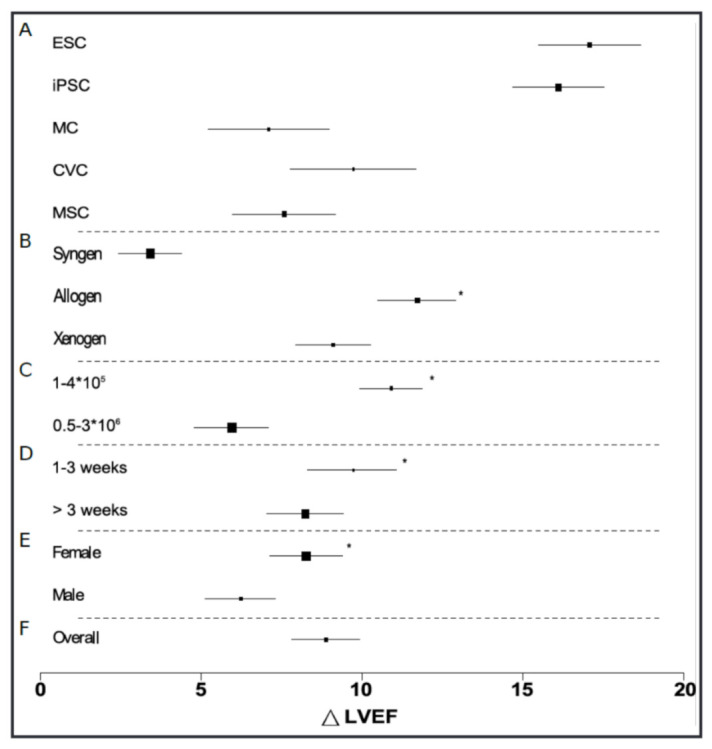
Subgroup analysis to identify significant moderators [Reprinted/adapted with permission from Ref. [66]. 2017, Cell Physiol Biochem Press]. Meta-regression analysis of subgroups revealed factors that significantly influence the magnitude of the functional improvement afforded by cell therapies—using LVEF improvement as a surrogate marker for efficacy. Cell type has no significant effect on the magnitude of LVEF improvement (*p* < 0.48; (**A**)). Cell origin has an impact on efficacy: allogeneic cells are most effective (12.9%; *p* = 0.046; (**B**)). Less than 500,000 cells are more effective than higher numbers (*p* = 0.013; (**C**)). The highest increase in LVEF can be measured up to 3 weeks post-transplantation (*p* = 0.004; (**D**)). Female mice benefit more from cardiac stem-cell therapies than male mice (*p* = 0.003; (**E**)). The overall effect of all investigated studies (**F**). * Marked as significant according to regression coefficient of the respective fixed-effects model.

**Figure 2 ijms-24-09017-f002:**
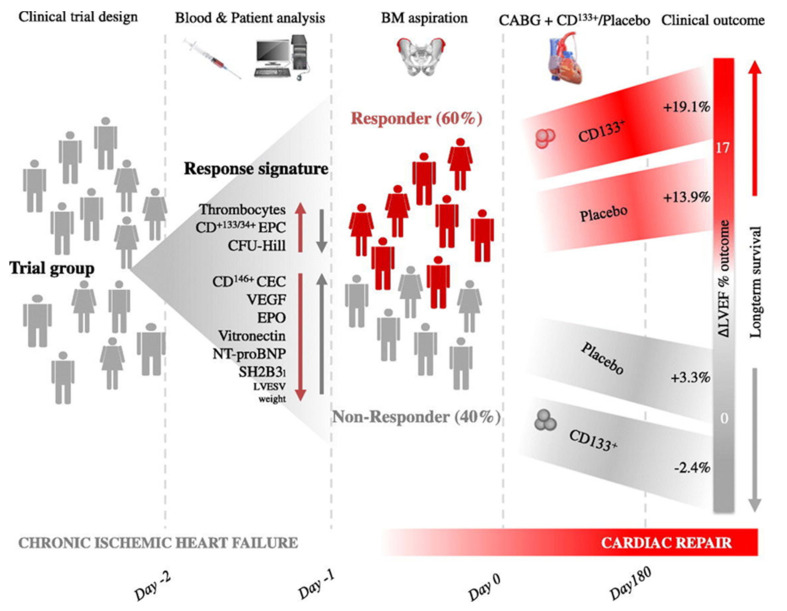
Cardiac Healing Modulation [Reprinted/adapted with permission from Ref. [72]. 2017, Elsevier]. The conclusion of our clinical studies on cardiac stem-cell research results in the following finding: The myocardial damage can be reduced by modulating the healing process, whereby patients are divided into responders (improvement in LVEF > 5% 180 days after bypass surgery) and non-responders. In addition, with non-responsiveness, angiogenesis is disturbed by a dysfunctional bone marrow response.

**Figure 3 ijms-24-09017-f003:**
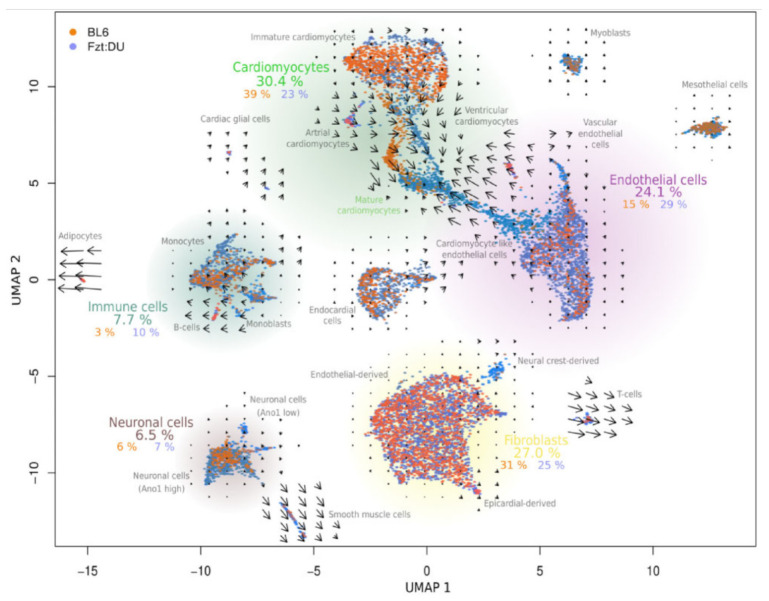
Single-nuclei transcriptome characteristics of inbred and outbred mouse hearts [Reprinted/adapted with permission from Ref. [96]. 2020, Oxford University Press]. UMAP clustering of snRNA-seq data reveals 23 distinct clusters for the indicated cell types. Percentages reflect the number of nuclei in the inbred mouse strain C57BL/6, the outbred strain Fzt:DU, and combined, respectively. The arrows represent RNA velocity kinetics, visualizing the direction and acceleration between mature and nascent mRNA. Due to binucleation in cardiomyocytes, the actual cellular content is lower (20% C57BL/6, 12% Fzt:DU).

## Data Availability

Not applicable.

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
