# Peer review of "New Approaches in Heart Research: Prevention Instead of Cardiomyoplasty?"

_ijms, 2023, doi:10.3390/ijms24109017_

Round 1
Reviewer 1 Report
This is a comprehensive report on the link between obesity, exercise, and cardiovascular disease which includes different perspectives from cell therapies and sports therapy to prevent cardiac diseases. This report could benefit from a section on the available specific treatment used in obese/diabetic patients for the primary and secondary prevention of MI, since it is known that for example statins and antidiabetic drugs have effects on adipogenesis.
Reviewer 2 Report
The present version of the paper seems to be suitable for the publication
Reviewer 3 Report
Firstly, the purpose of this review is not clear, general well-known theoretical data are described, which do not introduce anything new and which do not represent a scientific synthesis or systematic study.
Secondly, already published figures are used in the manuscript, the reason for this use in the original review is not clear.
Thirdly, a huge number of self-citations are present in this manuscript, one of the authors has 17 self-citations...
Minor editing of English language required
Round 2
Reviewer 3 Report
I agree with the changes made to the manuscript, but I remain of the opinion that the original article should contain the original figures.
Round 3
Reviewer 3 Report
The authors did not make the requested changes. Accept in present form, if the editors agree.